# Refined Lower Bounds for Adversarial Bandits

**Sébastien Gerchinovitz**
Institut de Mathématiques de Toulouse
Université Toulouse 3 Paul Sabatier
Toulouse, 31062, France
sebastien.gerchinovitz@math.univ-toulouse.fr

**Tor Lattimore**
Department of Computing Science
University of Alberta
Edmonton, Canada
tor.lattimore@gmail.com

## Abstract

We provide new lower bounds on the regret that must be suffered by adversarial bandit algorithms. The new results show that recent upper bounds that either (a) hold with high-probability or (b) depend on the total loss of the best arm or (c) depend on the quadratic variation of the losses, are close to tight. Besides this we prove two impossibility results. First, the existence of a single arm that is optimal in every round cannot improve the regret in the worst case. Second, the regret cannot scale with the effective range of the losses. In contrast, both results are possible in the full-information setting.

## 1  Introduction

We consider the standard $K$-armed adversarial bandit problem, which is a game played over $T$ rounds between a learner and an adversary. In every round $t \in \{1, \ldots, T\}$ the learner chooses a probability distribution $p_t = (p_{i,t})_{1 \leqslant i \leqslant K}$ over $\{1, \ldots, K\}$. The adversary then chooses a loss vector $\ell_t = (\ell_{i,t})_{1 \leqslant i \leqslant K} \in [0, 1]^K$, which may depend on $p_t$. Finally the learner samples an action from $p_t$ denoted by $I_t \in \{1, \ldots, K\}$ and observes her own loss $\ell_{I_t, t}$. The learner would like to minimise her regret, which is the difference between cumulative loss suffered and the loss suffered by the optimal action in hindsight:

$$R_T(\ell_{1:T}) = \sum_{t=1}^{T} \ell_{I_t, t} - \min_{1 \leqslant i \leqslant K} \sum_{t=1}^{T} \ell_{i,t} \,,$$

where $\ell_{1:T} \in [0, 1]^{TK}$ is the sequence of losses chosen by the adversary. A famous strategy is called Exp3, which satisfies $\mathbb{E}[R_T(\ell_{1:T})] = \mathcal{O}(\sqrt{KT \log(K)})$ where the expectation is taken over the randomness in the algorithm and the choices of the adversary [Auer et al., 2002]. There is also a lower bound showing that for every learner there is an adversary for which the expected regret is $\mathbb{E}[R_T(\ell_{1:T})] = \Omega(\sqrt{KT})$ [Auer et al., 1995]. If the losses are chosen ahead of time, then the adversary is called oblivious, and in this case there exists a learner for which $\mathbb{E}[R_T(\ell_{1:T})] = \mathcal{O}(\sqrt{KT})$ [Audibert and Bubeck, 2009]. One might think that this is the end of the story, but it is not so. While the worst-case expected regret is one quantity of interest, there are many situations where a refined regret guarantee is more informative. Recent research on adversarial bandits has primarily focussed on these issues, especially the questions of obtaining regret guarantees that hold with high probability as well as stronger guarantees when the losses are "nice" in some sense. While there are now a wide range of strategies with upper bounds that depend on various quantities, the literature is missing lower bounds for many cases, some of which we now provide.

We focus on three classes of lower bound, which are described in detail below. The first addresses the optimal regret achievable with high probability, where we show there is little room for improvement over existing strategies. Our other results concern lower bounds that depend on some kind of regularity in the losses ("nice" data). Specifically we prove lower bounds that replace $T$ in the regret bound with the loss of the best action (called first-order bounds) and also with the quadratic variation of the losses (called second-order bounds).

**High-probability bounds**   Existing strategies Exp3.P [Auer et al., 2002] and Exp3-IX [Neu, 2015a] are tuned with a confidence parameter $\delta \in (0,1)$ and satisfy, for all $\ell_{1:T} \in [0,1]^{KT}$,

$$\mathbb{P}\left(R_T(\ell_{1:T}) \geqslant c\sqrt{KT\log(K/\delta)}\right) \leqslant \delta \tag{1}$$

for some universal constant $c > 0$. An alternative tuning of Exp-IX or Exp3.P [Bubeck and Cesa-Bianchi, 2012] leads to a single algorithm for which, for all $\ell_{1:T} \in [0,1]^{KT}$,

$$\forall \delta \in (0,1) \qquad \mathbb{P}\left(R_T(\ell_{1:T}) \geqslant c\sqrt{KT}\left(\sqrt{\log(K)} + \frac{\log(1/\delta)}{\sqrt{\log(K)}}\right)\right) \leqslant \delta. \tag{2}$$

The difference is that in (1) the algorithm depends on $\delta$ while in (2) it does not. The cost of not knowing $\delta$ is that the $\log(1/\delta)$ moves outside the square root. In Section 2 we prove two lower bounds showing that there is little room for improvement in either (1) or (2).

**First-order bounds**   An improvement over the worst-case regret bound of $\mathcal{O}(\sqrt{TK})$ is the so-called *improvement for small losses*. Specifically, there exist strategies (eg., FPL-TRIX by Neu [2015b] with earlier results by Stoltz [2005], Allenberg et al. [2006], Rakhlin and Sridharan [2013]) such that for all $\ell_{1:T} \in [0,1]^{KT}$

$$\mathbb{E}[R_T(\ell_{1:T})] \leqslant \mathcal{O}\left(\sqrt{L_T^* K \log(K)} + K \log(KT)\right), \quad \text{with } L_T^* = \min_{1\leqslant i \leqslant K}\sum_{t=1}^{T}\ell_{i,t}, \tag{3}$$

where the expectation is with respect to the internal randomisation of the algorithm (the losses are fixed). This result improves on the $\mathcal{O}(\sqrt{KT})$ bounds since $L_T^* \leqslant T$ is always guaranteed and sometimes $L_T^*$ is much smaller than $T$. In order to evaluate the optimality of this bound, we first rewrite it in terms of the small-loss balls $\mathcal{B}_{\alpha,T}$ defined for all $\alpha \in [0,1]$ and $T \geqslant 1$ by

$$\mathcal{B}_{\alpha,T} \triangleq \left\{\ell_{1:T} \in [0,1]^{KT} : \frac{L_T^*}{T} \leqslant \alpha\right\}. \tag{4}$$

**Corollary 1.** *The first-order regret bound* (3) *of Neu [2015b] is equivalent to:*

$$\forall \alpha \in [0,1], \qquad \sup_{\ell_{1:T} \in \mathcal{B}_{\alpha,T}} \mathbb{E}[R_T(\ell_{1:T})] \leqslant \mathcal{O}\left(\sqrt{\alpha TK \log(K)} + K \log(KT)\right).$$

The proof is straightforward. Our main contribution in Section 3 is a lower bound of the order of $\sqrt{\alpha TK}$ for all $\alpha \in \Omega(\log(T)/T)$. This minimax lower bound shows that we cannot hope for a better bound than (3) (up to log factors) if we only know the value of $L_T^*$.

**Second-order bounds**   Another type of improved regret bound was derived by Hazan and Kale [2011b] and involves a second-order quantity called the quadratic variation:

$$Q_T = \sum_{t=1}^{T}\|\ell_t - \mu_T\|_2^2 \leqslant \frac{TK}{4}, \tag{5}$$

where $\mu_T = \frac{1}{T}\sum_{t=1}^{T}\ell_t$ is the mean of all loss vectors. (In other words, $Q_T/T$ is the sum of the empirical variances of all the $K$ arms). Hazan and Kale [2011b] addressed the general online linear optimisation setting. In the particular case of adversarial $K$-armed bandits with an oblivious adversary (as is the case here), they showed that there exists an efficient algorithm such that for some absolute constant $c > 0$ and for all $T \geqslant 2$

$$\forall \ell_{1:T} \in [0,1]^{KT}, \qquad \mathbb{E}[R_T(\ell_{1:T})] \leqslant c\left(K^2\sqrt{Q_T \log T} + K^{1.5}\log^2 T + K^{2.5}\log T\right). \tag{6}$$

As before we can rewrite the regret bound (6) in terms of the small-variation balls $\mathcal{V}_{\alpha,T}$ defined for all $\alpha \in [0,1/4]$ and $T \geqslant 1$ by

$$\mathcal{V}_{\alpha,T} \triangleq \left\{\ell_{1:T} \in [0,1]^{KT} : \frac{Q_T}{TK} \leqslant \alpha\right\}. \tag{7}$$

**Corollary 2.** *The second-order regret bound* (6) *of Hazan and Kale [2011b] is equivalent to:*

$$\forall \alpha \in [0,1/4], \qquad \sup_{\ell_{1:T} \in \mathcal{V}_{\alpha,T}} \mathbb{E}[R_T(\ell_{1:T})] \leqslant c\left(K^2\sqrt{\alpha TK \log T} + K^{3/2}\log^2 T + K^{5/2}\log T\right).$$

The proof is straightforward because the losses are deterministic and fixed in advance by an oblivious adversary. In Section 4 we provide a lower bound of order $\sqrt{\alpha TK}$ that holds whenever $\alpha = \Omega(\log(T)/T)$. This minimax lower bound shows that we cannot hope for a bound better than (7) by more than a factor of $K^2\sqrt{\log T}$ if we only know the value of $Q_T$. Closing the gap is left as an open question.

**Two impossibility results in the bandit setting**   We also show in Section 4 that, in contrast to the full-information setting, regret bounds involving the cumulative variance of the algorithm as in [Cesa-Bianchi et al., 2007] cannot be obtained in the bandit setting. More precisely, we prove that two consequences that hold true in the full-information case, namely: (i) a regret bound proportional to the effective range of the losses and (ii) a bounded regret if one arm performs best at all rounds, must fail in the worst case for every bandit algorithm.

**Additional notation and key tools**   Before the theorems we develop some additional notation and describe the generic ideas in the proofs. For $1 \leqslant i \leqslant K$ let $N_i(t)$ be the number of times action $i$ has been chosen after round $t$. All our lower bounds are derived by analysing the regret incurred by strategies when facing randomised adversaries that choose the losses for all actions from the same joint distribution in every round (sometimes independently for each action and sometimes not). $\mathrm{Ber}(\alpha)$ denotes the Bernoulli distribution with parameter $\alpha \in [0,1]$. If $\mathbb{P}$ and $\mathbb{Q}$ are measures on the same probability space, then $\mathrm{KL}(\mathbb{P}, \mathbb{Q})$ is the KL-divergence between them. For $a < b$ we define $\mathrm{clip}_{[a,b]}(x) = \min\{b, \max\{a, x\}\}$ and for $x, y \in \mathbb{R}$ we let $x \vee y = \max\{x, y\}$. Our main tools throughout the analysis are the following information-theoretic lemmas. The first bounds the KL divergence between the laws of the observed losses/actions for two distributions on the losses.

**Lemma 1.** *Fix a randomised bandit algorithm and two probability distributions $Q_1$ and $Q_2$ on $[0,1]^K$. Assume the loss vectors $\ell_1, \ldots, \ell_T \in [0,1]^K$ are drawn i.i.d. from either $Q_1$ or $Q_2$, and denote by $\mathbb{Q}_j$ the joint probability distribution on all sources of randomness when $Q_j$ is used (formally, $\mathbb{Q}_j = \mathbb{P}_{\mathrm{int}} \otimes (Q_j^{\otimes T})$, where $\mathbb{P}_{\mathrm{int}}$ is the probability distribution used by the algorithm for its internal randomisation). Let $t \geqslant 1$. Denote by $h_t = (I_s, \ell_{I_s,s})_{1 \leqslant s \leqslant t-1}$ the history available at the beginning of round $t$, by $\mathbb{Q}_j^{(h_t, I_t)}$ the law of $(h_t, I_t)$ under $\mathbb{Q}_j$, and by $Q_{j,i}$ the $i$th marginal distribution of $Q_j$. Then,*

$$\mathrm{KL}\left(\mathbb{Q}_1^{(h_t, I_t)}, \mathbb{Q}_2^{(h_t, I_t)}\right) = \sum_{i=1}^{K} \mathbb{E}_{\mathbb{Q}_1}\left[N_i(t-1)\right] \mathrm{KL}\left(Q_{1,i}, Q_{2,i}\right) .$$

Results of roughly this form are well known and the proof follows immediately from the chain rule for the relative entropy and the independence of the loss vectors across time (see [Auer et al., 2002] or the supplementary material). One difference is that the losses need not be independent across the arms, which we heavily exploit in our proofs by using correlated losses. The second key lemma is an alternative to Pinsker's inequality that proves useful when the Kullback-Leibler divergence is larger than 2. It has previously been used for bandit lower bounds (in the stochastic setting) by Bubeck et al. [2013].

**Lemma 2** (Lemma 2.6 in Tsybakov 2008). *Let $P$ and $Q$ be two probability distributions on the same measurable space. Then, for every measurable subset $A$ (whose complement we denote by $A^c$),*

$$P(A) + Q(A^c) \geqslant \frac{1}{2} \exp\left(-\mathrm{KL}(P, Q)\right) .$$

## 2   Zero-Order High Probability Lower Bounds

We prove two new high-probability lower bounds on the regret of any bandit algorithm. The first shows that no strategy can enjoy smaller regret than $\Omega(\sqrt{KT \log(1/\delta)})$ with probability at least $1 - \delta$. Upper bounds of this form have been shown for various algorithms including Exp3.P [Auer et al., 2002] and Exp3-IX [Neu, 2015a]. Although this result is not very surprising, we are not aware of any existing work on this problem and the proof is less straightforward than one might expect. An added benefit of our result is that the loss sequences producing large regret have two special properties. First, the optimal arm is the same in every round and second the range of the losses in each round is $\mathcal{O}(\sqrt{K \log(1/\delta)/T})$. These properties will be useful in subsequent analysis.

In the second lower bound we show that any algorithm for which $\mathbb{E}[R_T(\ell_{1:T})] = \mathcal{O}(\sqrt{KT})$ must necessarily suffer a high probability regret of at least $\Omega(\sqrt{KT \log(1/\delta)})$ for some sequence $\ell_{1:T}$.

The important difference relative to the previous result is that strategies with $\log(1/\delta)$ appearing inside the square root depend on a specific value of $\delta$, which must be known in advance.

**Theorem 1.** *Suppose $K \geqslant 2$ and $\delta \in (0, 1/4)$ and $T \geqslant 32(K-1)\log(2/\delta)$, then there exists a sequence of losses $\ell_{1:T} \in [0,1]^{KT}$ such that*

$$\mathbb{P}\left(R_T(\ell_{1:T}) \geqslant \frac{1}{27}\sqrt{(K-1)T\log(1/(4\delta))}\right) \geqslant \delta/2\,,$$

*where the probability is taken with respect to the randomness in the algorithm. Furthermore $\ell_{1:T}$ can be chosen in such a way that there exists an $i$ such that for all $t$ it holds that $\ell_{i,t} = \min_j \ell_{j,t}$ and $\max_{j,k}\{\ell_{j,t} - \ell_{k,t}\} \leqslant \sqrt{(K-1)\log(1/(4\delta))/T}/(4\sqrt{\log 2})$.*

**Theorem 2.** *Suppose $K \geqslant 2$, $T \geqslant 1$, and there exists a strategy and constant $C > 0$ such that for any $\ell_{1:T} \in [0,1]^{KT}$ it holds that $\mathbb{E}[R_T(\ell_{1:T})] \leqslant C\sqrt{(K-1)T}$. Let $\delta \in (0, 1/4)$ satisfy $\sqrt{(K-1)/T}\log(1/(4\delta)) \leqslant C$ and $T \geqslant 32\log(2/\delta)$. Then there exists $\ell_{1:T} \in [0,1]^{KT}$ for which*

$$\mathbb{P}\left(R_T(\ell_{1:T}) \geqslant \frac{\sqrt{(K-1)T}\log(1/(4\delta))}{203C}\right) \geqslant \delta/2\,,$$

*where the probability is taken with respect to the randomness in the algorithm.*

**Corollary 3.** *If $p \in (0,1)$ and $C > 0$, then there does not exist a strategy such that for all $T$, $K$, $\ell_{1:T} \in [0,1]^{KT}$ and $\delta \in (0,1)$ the regret is bounded by $\mathbb{P}\left(R_T(\ell_{1:T}) \geqslant C\sqrt{(K-1)T}\log^p(1/\delta)\right) \leqslant \delta$.*

The corollary follows easily by integrating the assumed high-probability bound and applying Theorem 2 for sufficiently large $T$ and small $\delta$. The proof may be found in the supplementary material.

**Proof of Theorems 1 and 2**    Both proofs rely on a carefully selected choice of correlated stochastic losses described below. Let $Z_1, Z_2, \ldots, Z_T$ be a sequence of i.i.d. Gaussian random variables with mean $1/2$ and variance $\sigma^2 = 1/(32\log(2))$. Let $\Delta \in [0, 1/30]$ be a constant that will be chosen differently in each proof and define $K$ random loss sequences $\ell_{1:T}^1, \ldots, \ell_{1:T}^K$ where

$$\ell_{i,t}^j = \begin{cases} \text{clip}_{[0,1]}(Z_t - \Delta) & \text{if } i = 1 \\ \text{clip}_{[0,1]}(Z_t - 2\Delta) & \text{if } i = j \neq 1 \\ \text{clip}_{[0,1]}(Z_t) & \text{otherwise}\,. \end{cases}$$

For $1 \leqslant j \leqslant K$ let $\mathbb{Q}_j$ be the measure on $\ell_{1:T} \in [0,1]^{KT}$ and $I_1, \ldots, I_T$ when $\ell_{i,t} = \ell_{i,t}^j$ for all $1 \leqslant i \leqslant K$ and $1 \leqslant t \leqslant T$. Informally, $\mathbb{Q}_j$ is the measure on the sequence of loss vectors and actions when the learner interacts with the losses sampled from the $j$th environment defined above.

**Lemma 3.** *Let $\delta \in (0,1)$ and suppose $\Delta \leqslant 1/30$ and $T \geqslant 32\log(2/\delta)$. Then $\mathbb{Q}_i\left(R_T(\ell_{1:T}^i) \geqslant \Delta T/4\right) \geqslant \mathbb{Q}_i\left(N_i(T) \leqslant T/2\right) - \delta/2$ and $\mathbb{E}_{\mathbb{Q}_i}[R_T(\ell_{1:T}^i)] \geqslant 7\Delta\mathbb{E}_{\mathbb{Q}_i}[T - N_i(T)]/8$.*

The proof follows by substituting the definition of the losses and applying Azuma's inequality to show that clipping does not occur too often. See the supplementary material for details.

*Proof of Theorem 1.* First we choose the value of $\Delta$ that determines the gaps in the losses by $\Delta = \sqrt{\sigma^2(K-1)\log(1/(4\delta))/(2T)} \leqslant 1/30$. By the pigeonhole principle there exists an $i > 1$ for which $\mathbb{E}_{\mathbb{Q}_1}[N_i(T)] \leqslant T/(K-1)$. Therefore by Lemmas 2 and 1, and the fact that the KL divergence between clipped Gaussian distributions is always smaller than without clipping,

$$\mathbb{Q}_1\left(N_1(T) \leqslant T/2\right) + \mathbb{Q}_i\left(N_1(T) > T/2\right) \geqslant \frac{1}{2}\exp\left(-\text{KL}\left(\mathbb{Q}_1^{(h_T, I_T)}, \mathbb{Q}_i^{(h_T, I_T)}\right)\right)$$

$$\geqslant \frac{1}{2}\exp\left(-\frac{\mathbb{E}_{\mathbb{Q}_1}[N_i(T)](2\Delta)^2}{2\sigma^2}\right) \geqslant \frac{1}{2}\exp\left(-\frac{2T\Delta^2}{\sigma^2(K-1)}\right) = 2\delta\,.$$

But by Lemma 3

$$\max_{k \in \{1,i\}} \mathbb{Q}_k\left(R_T(\ell_{1:T}^k) \geqslant T\Delta/4\right) \geqslant \max\left\{\mathbb{Q}_1\left(N_1(T) \leqslant T/2\right), \mathbb{Q}_i\left(N_i(T) \leqslant T/2\right)\right\} - \delta/2$$

$$\geqslant \frac{1}{2}\left(\mathbb{Q}_1\left(N_1(T) \leqslant T/2\right) + \mathbb{Q}_i\left(N_1(T) > T/2\right)\right) - \delta/2 \geqslant \delta/2\,.$$

Therefore there exists an $i \in \{1, \ldots, K\}$ such that

$$\mathbb{Q}_i \left( R_T(\ell^i_{1:T}) \geqslant \sqrt{\frac{\sigma^2 T(K-1)}{32} \log\left(\frac{1}{4\delta}\right)} \right) = \mathbb{Q}_i \left( R_T(\ell^i_{1:T}) \geqslant T\Delta/4 \right) \geqslant \delta/2 \,.$$

The result is completed by substituting the value of $\sigma^2 = 1/(32\log(2))$ and by noting that $\max_{j,k}\{\ell_{j,t} - \ell_{k,t}\} \leqslant 2\Delta \leqslant \sqrt{(K-1)\log(1/(4\delta))/T}/(4\sqrt{\log 2})$ $\mathbb{Q}_i$-almost surely. $\qquad\square$

*Proof of Theorem 2.* By the assumption on $\delta$ we have $\Delta = \frac{7\sigma^2}{16C}\sqrt{\frac{K-1}{T}\log\left(\frac{1}{4\delta}\right)} \leqslant 1/30$. Suppose for all $i > 1$ that

$$\mathbb{E}_{\mathbb{Q}_1}[N_i(T)] > \frac{\sigma^2}{2\Delta^2}\log\left(\frac{1}{4\delta}\right) \,. \tag{8}$$

Then by the assumption in the theorem statement and the second part of Lemma 3 we have

$$C\sqrt{(K-1)T} \geqslant \mathbb{E}_{\mathbb{Q}_1}[R_T(\ell^1_{1:T})] \geqslant \frac{7\Delta}{8}\mathbb{E}_{\mathbb{Q}_1}\left[\sum_{i=2}^{K} N_i(T)\right] > \frac{7\sigma^2(K-1)}{16\Delta}\log\frac{1}{4\delta} = C\sqrt{(K-1)T}\,,$$

which is a contradiction. Therefore there exists an $i > 1$ for which Eq. (8) does not hold. Then by the same argument as the previous proof it follows that

$$\max_{k\in\{1,i\}} \mathbb{Q}_k\left(R_T(\ell^k_{1:T}) \geqslant \frac{7\sigma^2}{4\cdot16C}\sqrt{(K-1)T}\log\frac{1}{4\delta}\right) = \max_{k\in\{1,i\}} \mathbb{Q}_k\left(R_T(\ell^k_{1:T}) \geqslant T\Delta/4\right) \geqslant \delta/2\,.$$

The result is completed by substituting the value of $\sigma^2 = 1/(32\log(2))$. $\qquad\square$

**Remark 1.** *It is possible to derive similar high-probability regret bounds with non-correlated losses. However the correlation makes the results cleaner (we do not need an additional concentration argument to locate the optimal arm) and it is key to derive Corollaries 4 and 5 in Section 4.*

## 3 First-Order Lower Bound

First-order upper bounds provide improvement over minimax bounds when the loss of the optimal action is small. Recall from Corollary 1 that first-order bounds can be rewritten in terms of the small-loss balls $\mathcal{B}_{\alpha,T}$ defined in (4). Theorem 3 below provides a new lower bound of order $\sqrt{L_T^* K}$, which matches the best existing upper bounds up to logarithmic factors. As is standard for minimax results this does not imply a lower bound on every loss sequence $\ell_{1:T}$. Instead it shows that we cannot hope for a better bound if we only know the value of $L_T^*$.

**Theorem 3.** *Let $K \geqslant 2$, $T \geqslant K \vee 118$, and $\alpha \in [(c\log(32T) \vee (K/2))/T, 1/2]$, where $c = 64/9$. Then for any randomised bandit algorithm $\sup_{\ell_{1:T} \in \mathcal{B}_{\alpha,T}} \mathbb{E}[R_T(\ell_{1:T})] \geqslant \sqrt{\alpha T K}/27$, where the expectation is taken with respect to the internal randomisation of the algorithm.*

Our proof is inspired by that of Auer et al. [2002, Theorem 5.1]. The key difference is that we take Bernoulli distributions with parameter close to $\alpha$ instead of $1/2$. This way the best cumulative loss $L_T^*$ is ensured to be concentrated around $\alpha T$, and the regret lower bound $\sqrt{\alpha T K} \approx \sqrt{\alpha(1-\alpha)T K}$ can be seen to involve the variance $\alpha(1-\alpha)T$ of the binomial distribution with parameters $\alpha$ and $T$.

First we state the stochastic construction of the losses and prove a general lemma that allows us to prove Theorem 3 and will also be useful in Section 4 to a derive a lower bound in terms of the quadratic variation. Let $\varepsilon \in [0, 1-\alpha]$ be fixed and define $K$ probability distributions $(\mathbb{Q}_j)_{j=1}^K$ on $[0,1]^{KT}$ such that under $\mathbb{Q}_j$ the following hold:

- All random losses $\ell_{i,t}$ for $1 \leqslant i \leqslant K$ and $1 \leqslant t \leqslant T$ are independent.
- $\ell_{i,t}$ is sampled from a Bernoulli distribution with parameter $\alpha + \varepsilon$ if $i \neq j$, or with parameter $\alpha$ if $i = j$.

**Lemma 4.** *Let $\alpha \in (0,1)$, $K \geqslant 2$, and $T \geqslant K/(4(1-\alpha))$. Consider the probability distributions $\mathbb{Q}_j$ on $[0,1]^{KT}$ defined above with $\varepsilon = (1/2)\sqrt{\alpha(1-\alpha)K/T}$, and set $\bar{\mathbb{Q}} = \frac{1}{K}\sum_{j=1}^K \mathbb{Q}_j$. Then for any randomised bandit algorithm $\mathbb{E}[R_T(\ell_{1:T})] \geqslant \sqrt{\alpha(1-\alpha)T K}/8$, where the expectation is with respect to both the internal randomisation of the algorithm and the random loss sequence $\ell_{1:T}$ which is drawn from $\bar{\mathbb{Q}}$.*

The assumption $T \geqslant K/(4(1-\alpha))$ above ensures that $\varepsilon \leqslant 1 - \alpha$, so that the $\mathbb{Q}_j$ are well defined.

*Proof of Lemma 4.* We lower bound the regret by the pseudo-regret for each distribution $\mathbb{Q}_j$:

$$
\mathbb{E}_{\mathbb{Q}_j}\left[\sum_{t=1}^{T} \ell_{I_t,t} - \min_{1 \leqslant i \leqslant K} \sum_{t=1}^{T} \ell_{i,t}\right] \geqslant \mathbb{E}_{\mathbb{Q}_j}\left[\sum_{t=1}^{T} \ell_{I_t,t}\right] - \min_{1 \leqslant i \leqslant K} \mathbb{E}_{\mathbb{Q}_j}\left[\sum_{t=1}^{T} \ell_{i,t}\right]
$$
$$
= \sum_{t=1}^{T} \mathbb{E}_{\mathbb{Q}_j}\left[\alpha + \varepsilon - \varepsilon \mathbb{1}_{\{I_t=j\}}\right] - T\alpha = T\varepsilon\left(1 - \frac{1}{T}\sum_{t=1}^{T} \mathbb{Q}_j(I_t = j)\right), \qquad (9)
$$

where the first equality follows because $\mathbb{E}_{\mathbb{Q}_j}[\ell_{I_t,t}] = \mathbb{E}_{\mathbb{Q}_j}[\mathbb{E}_{\mathbb{Q}_j}[\ell_{I_t,t}|\ell_{1:t-1},I_t]] = \mathbb{E}_{\mathbb{Q}_j}[\alpha + \varepsilon - \varepsilon \mathbb{1}_{\{I_t=j\}}]$ since under $\mathbb{Q}_j$, the conditional distribution of $\ell_t$ given $(\ell_{1:t-1}, I_t)$ is simply $\otimes_{i=1}^{K}\mathcal{B}(\alpha + \varepsilon - \varepsilon \mathbb{1}_{\{i=j\}})$. To bound (9) from below, note that by Pinsker's inequality we have for all $t \in \{1,\ldots,T\}$ and $j \in \{1,\ldots,K\}$, $\mathbb{Q}_j(I_t = j) \leqslant \mathbb{Q}_0(I_t = j) + (\mathrm{KL}(\mathbb{Q}_0^{I_t},\mathbb{Q}_j^{I_t})/2)^{1/2}$, where $\mathbb{Q}_0 = \mathrm{Ber}(\alpha + \varepsilon)^{\otimes KT}$ is the joint probability distribution that makes all the $\ell_{i,t}$ i.i.d. $\mathrm{Ber}(\alpha + \varepsilon)$, and $\mathbb{Q}_0^{I_t}$ and $\mathbb{Q}_j^{I_t}$ denote the laws of $I_t$ under $\mathbb{Q}_0$ and $\mathbb{Q}_j$ respectively. Plugging the last inequality above into (9), averaging over $j = 1,\ldots,K$ and using the concavity of the square root yields

$$
\mathbb{E}_{\bar{\mathbb{Q}}}\left[\sum_{t=1}^{T} \ell_{I_t,t} - \min_{1 \leqslant i \leqslant K} \sum_{t=1}^{T} \ell_{i,t}\right] \geqslant T\varepsilon\left(1 - \frac{1}{K} - \sqrt{\frac{1}{2T}\sum_{t=1}^{T}\frac{1}{K}\sum_{j=1}^{K}\mathrm{KL}(\mathbb{Q}_0^{I_t},\mathbb{Q}_j^{I_t})}\right), \qquad (10)
$$

where we recall that $\bar{\mathbb{Q}} = \frac{1}{K}\sum_{j=1}^{K}\mathbb{Q}_j$. The rest of the proof is devoted to upper-bounding $\mathrm{KL}(\mathbb{Q}_0^{I_t},\mathbb{Q}_j^{I_t})$. Denote by $h_t = (I_s, \ell_{I_s,s})_{1 \leqslant s \leqslant t-1}$ the history available at the beginning of round $t$. From Lemma 1

$$
\mathrm{KL}\left(\mathbb{Q}_0^{I_t},\mathbb{Q}_j^{I_t}\right) \leqslant \mathrm{KL}\left(\mathbb{Q}_0^{(h_t,I_t)},\mathbb{Q}_j^{(h_t,I_t)}\right) = \mathbb{E}_{\mathbb{Q}_0}\left[N_j(t-1)\right]\mathrm{KL}\left(\mathcal{B}(\alpha + \varepsilon),\mathcal{B}(\alpha)\right)
$$
$$
\leqslant \mathbb{E}_{\mathbb{Q}_0}\left[N_j(t-1)\right]\frac{\varepsilon^2}{\alpha(1-\alpha)}, \qquad (11)
$$

where the last inequality follows by upper bounding the KL divergence by the $\chi^2$ divergence (see the supplementary material). Averaging (11) over $j \in \{1,\ldots,K\}$ and $t \in \{1,\ldots,T\}$ and noting that $\sum_{t=1}^{T}(t-1) \leqslant T^2/2$ we get

$$
\frac{1}{T}\sum_{t=1}^{T}\frac{1}{K}\sum_{j=1}^{K}\mathrm{KL}(\mathbb{Q}_0^{I_t},\mathbb{Q}_j^{I_t}) \leqslant \frac{1}{T}\sum_{t=1}^{T}\frac{(t-1)\varepsilon^2}{K\alpha(1-\alpha)} \leqslant \frac{T\varepsilon^2}{2K\alpha(1-\alpha)}.
$$

Plugging the above inequality into (10) and using the definition of $\varepsilon = (1/2)\sqrt{\alpha(1-\alpha)K/T}$ yields

$$
\mathbb{E}_{\bar{\mathbb{Q}}}\left[\sum_{t=1}^{T} \ell_{I_t,t} - \min_{1 \leqslant i \leqslant K} \sum_{t=1}^{T} \ell_{i,t}\right] \geqslant T\varepsilon\left(1 - \frac{1}{K} - \frac{1}{4}\right) \geqslant \frac{1}{8}\sqrt{\alpha(1-\alpha)TK}. \qquad \square
$$

*Proof of Theorem 3.* We show that there exists a loss sequence $\ell_{1:T} \in [0,1]^{KT}$ such that $L_T^* \leqslant \alpha T$ and $\mathbb{E}[R_T(\ell_{1:T})] \geqslant (1/27)\sqrt{\alpha TK}$. Lemma 4 above provides such kind of lower bound, but without the guarantee on $L_T^*$. For this purpose we will use Lemma 4 with a smaller value of $\alpha$ (namely, $\alpha/2$) and combine it with Bernstein's inequality to prove that $L_T^* \leqslant T\alpha$ with high probability.

Part 1: Applying Lemma 4 with $\alpha/2$ (note that $T \geqslant K \geqslant K/(4(1-\alpha/2))$ by assumption on $T$) and noting that $\max_j \mathbb{E}_{\mathbb{Q}_j}[R_T(\ell_{1:T})] \geqslant \mathbb{E}_{\bar{\mathbb{Q}}}[R_T(\ell_{1:T})]$ we get that for some $j \in \{1,\ldots,K\}$ the probability distribution $\mathbb{Q}_j$ defined with $\varepsilon = (1/2)\sqrt{(\alpha/2)(1-\alpha/2)K/T}$ satisfies

$$
\mathbb{E}_{\mathbb{Q}_j}[R_T(\ell_{1:T})] \geqslant \frac{1}{8}\sqrt{\frac{\alpha}{2}\left(1 - \frac{\alpha}{2}\right)TK} \geqslant \frac{1}{32}\sqrt{6\alpha TK} \qquad (12)
$$

since $\alpha \leqslant 1/2$ by assumption.

Part 2: Next we prove that $\qquad \mathbb{Q}_j(L_T^* > T\alpha) \leqslant \frac{1}{32T}. \qquad (13)$

To this end, first note that $L_T^* \leqslant \sum_{t=1}^T \ell_{j,t}$. Second, note that under $\mathbb{Q}_j$, the $\ell_{j,t}$, $t \geqslant 1$, are i.i.d. $\mathrm{Ber}(\alpha/2)$. We can thus use Bernstein's inequality: applying Theorem 2.10 (and a remark on p.38) of Boucheron et al. [2013] with $X_t = \ell_{j,t} - \alpha/2 \leqslant 1 = b$, with $v = T(\alpha/2)(1 - \alpha/2)$, and with $c = b/3 = 1/3$), we get that, for all $\delta \in (0,1)$, with $\mathbb{Q}_j$-probability at least $1 - \delta$,

$$L_T^* \leqslant \sum_{t=1}^T \ell_{j,t} \leqslant \frac{T\alpha}{2} + \sqrt{2T \frac{\alpha}{2}\left(1 - \frac{\alpha}{2}\right)\log\frac{1}{\delta}} + \frac{1}{3}\log\frac{1}{\delta}$$

$$\leqslant \frac{T\alpha}{2} + \left(1 + \frac{1}{3}\right)\sqrt{T\alpha \log\frac{1}{\delta}} \leqslant \frac{T\alpha}{2} + \frac{T\alpha}{2} = T\alpha , \tag{14}$$

where the second last inequality is true whenever $T\alpha \geqslant \log(1/\delta)$ and that last is true whenever $T\alpha \geqslant (8/3)^2 \log(1/\delta) = c\log(1/\delta)$. By assumption on $\alpha$, these two conditions are satisfied for $\delta = 1/(32T)$, which concludes the proof of (13).

Conclusion: We show by contradiction that there exists a loss sequence $\ell_{1:T} \in [0,1]^{KT}$ such that $L_T^* \leqslant \alpha T$ and

$$\mathbb{E}[R_T(\ell_{1:T})] \geqslant \frac{1}{64}\sqrt{6\alpha TK} , \tag{15}$$

where the expectation is with respect to the internal randomisation of the algorithm. Imagine for a second that (15) were false for every loss sequence $\ell_{1:T} \in [0,1]^{KT}$ satisfying $L_T^* \leqslant \alpha T$. Then we would have $\mathbb{1}_{\{L_T^* \leqslant \alpha T\}}\mathbb{E}_{\mathbb{Q}_j}[R_T(\ell_{1:T})|\ell_{1:T}] \leqslant (1/64)\sqrt{6\alpha TK}$ almost surely (since the internal source of randomness of the bandit algorithm is independent of $\ell_{1:T}$). Therefore by the tower rule for the first expectation on the r.h.s. below, we would get

$$\mathbb{E}_{\mathbb{Q}_j}[R_T(\ell_{1:T})] = \mathbb{E}_{\mathbb{Q}_j}\left[R_T(\ell_{1:T})\mathbb{1}_{\{L_T^* \leqslant \alpha T\}}\right] + \mathbb{E}_{\mathbb{Q}_j}\left[R_T(\ell_{1:T})\mathbb{1}_{\{L_T^* > \alpha T\}}\right]$$

$$\leqslant \frac{1}{64}\sqrt{6\alpha TK} + T \cdot \mathbb{Q}_j(L_T^* > T\alpha) \leqslant \frac{1}{64}\sqrt{6\alpha TK} + \frac{1}{32} < \frac{1}{32}\sqrt{6\alpha TK} \tag{16}$$

where (16) follows from (13) and by noting that $1/32 < (1/64)\sqrt{6\alpha TK}$ since $\alpha \geqslant K/(2T) > 4/(6T) \geqslant 4/(6TK)$. Comparing (16) and (12) we get a contradiction, which proves that there exists a loss sequence $\ell_{1:T} \in [0,1]^{KT}$ satisfying both $L_T^* \leqslant \alpha T$ and (15). We conclude the proof by noting that $\sqrt{6}/64 \geqslant 1/27$. Finally, the condition $T \geqslant K \vee 118$ is sufficient to make the interval $\left[(c\log(32T) \vee (K/2))/T, \frac{1}{2}\right]$ non empty. $\square$

## 4   Second-Order Lower Bounds

We start by giving a lower bound on the regret in terms of the quadratic variation that is close to existing upper bounds except in the dependence on the number of arms. Afterwards we prove that bandit strategies cannot adapt to losses that lie in a small range or the existence of an action that is always optimal.

**Lower bound in terms of quadratic variation**   We prove a lower bound of $\Omega(\sqrt{\alpha TK})$ over any small-variation ball $\mathcal{V}_{\alpha,T}$ (as defined by (7)) for all $\alpha = \Omega(\log(T)/T)$. This minimax lower bound matches the upper bound of Corollary 2 up to a multiplicative factor of $K^2\sqrt{\log(T)}$. Closing this gap is left as an open question, but we conjecture that the upper bound is loose (see also the COLT open problem by Hazan and Kale [2011a]).

**Theorem 4.** *Let $K \geqslant 2$, $T \geqslant (32K) \vee 601$, and $\alpha \in [(2c_1 \log(T) \vee 8K)/T, 1/4]$, where $c_1 = (4/9)^2(3\sqrt{5} + 1)^2 \leqslant 12$. Then for any randomised bandit algorithm, $\sup_{\ell_{1:T} \in \mathcal{V}_{\alpha,T}} \mathbb{E}[R_T(\ell_{1:T})] \geqslant \sqrt{\alpha TK}/25$, where the expectation is taken with respect to the internal randomisation of the algorithm.*

The proof is very similar to that of Theorem 3; it also follows from Lemma 4 and Bernstein's inequality. It is postponed to the supplementary material.

**Impossibility results**   In the full-information setting (where the entire loss vector is observed after each round) Cesa-Bianchi et al. [2007, Theorem 6] designed a carefully tuned exponential weighting algorithm for which the regret depends on the variation of the algorithm and the range of the losses:

$$\forall \ell_{1:T} \in \mathbb{R}^{KT}, \qquad \mathbb{E}[R_T(\ell_{1:T})] \leqslant 4\sqrt{V_T \log K} + 4E_T \log K + 6E_T , \tag{17}$$

where the expectation is taken with respect to the internal randomisation of the algorithm and $E_T = \max_{1 \leqslant t \leqslant T} \max_{1 \leqslant i,j \leqslant K} |\ell_{i,t} - \ell_{j,t}|$ denotes the effective range of the losses and $V_T = \sum_{t=1}^T \mathrm{Var}_{I_t \sim p_t}(\ell_{I_t,t})$ denotes the cumulative variance of the algorithm (in each round $t$ the expert's action $I_t$ is drawn at random from the weight vector $p_t$). The bound in (17) is not closed-form because $V_T$ depends on the algorithm, but has several interesting consequences:

1. If for all $t$ the losses $\ell_{i,t}$ lie in an unknown interval $[a_t, a_t + \rho]$ with a small width $\rho > 0$, then $\mathrm{Var}_{I_t \sim p_t}(\ell_{I_t,t}) \leqslant \rho^2/4$, so that $V_T \leqslant T\rho^2/4$. Hence

$$\mathbb{E}[R_T(\ell_{1:T})] \leqslant 2\rho\sqrt{T \log K} + 4\rho \log K + 6\rho \ .$$

   Therefore, though the algorithm by Cesa-Bianchi et al. [2007, Section 4.2] does not use the prior knowledge of $a_t$ or $\rho$, it is able to incur a regret that scales linearly in the effective range $\rho$.
2. If all the losses $\ell_{i,t}$ are nonnegative, then by Corollary 3 of [Cesa-Bianchi et al., 2007] the second-order bound (17) implies the first-order bound

$$\mathbb{E}[R_T(\ell_{1:T})] \leqslant 4\sqrt{L_T^*\left(M_T - \frac{L_T^*}{T}\right)\log K} + 39 M_T \max\{1, \log K\} \ , \qquad (18)$$

   where $M_T = \max_{1 \leqslant t \leqslant T} \max_{1 \leqslant i \leqslant K} \ell_{i,t}$ .
3. If there exists an arm $i^*$ that is optimal at every round $t$ (i.e., $\ell_{i^*,t} = \min_i \ell_{i,t}$ for all $t \geqslant 1$), then any translation-invariant algorithm with regret guarantees as in (18) above suffers a bounded regret. This is the case for the fully automatic algorithm of Cesa-Bianchi et al. [2007, Theorem 6] mentioned above. Then by the translation invariance of the algorithm all losses $\ell_{i,t}$ appearing in the regret bound can be replaced with the translated losses $\ell_{i,t} - \ell_{i^*,t} \geqslant 0$, so that a bound of the same form as (18) implies a regret bound of $\mathcal{O}(\log K)$.
4. Assume that the loss vectors $\ell_t$ are i.i.d. with a unique optimal arm in expectation (i.e., there exists $i^*$ such that $\mathbb{E}[\ell_{i^*,1}] < \mathbb{E}[\ell_{i,1}]$ for all $i \neq i^*$). Then using the Hoeffding-Azuma inequality we can show that the algorithm of Cesa-Bianchi et al. [2007, Section 4.2] has with high probability a bounded cumulative variance $V_T$, and therefore (by (17)) incurs a bounded regret, in the same spirit as in de Rooij et al. [2014], Gaillard et al. [2014].

We already know that point 2 has a counterpart in the bandit setting. If one is prepared to ignore logarithmic terms, then point 4 also has an analogue in the bandit setting due to the existence of logarithmic regret guarantees for stochastic bandits [Lai and Robbins, 1985]. The following corollaries show that in the bandit setting it is not possible to design algorithms to exploit the range of the losses or the existence of an arm that is always optimal. We use Theorem 1 as a general tool but the bounds can be improved to $\sqrt{TK}/30$ by analysing the expected regret directly (similar to Lemma 4).

**Corollary 4.** *Let $K \geqslant 2$, $T \geqslant 32(K-1)\log(14)$ and $\rho \geqslant 0.22\sqrt{(K-1)/T}$. Then for any randomised bandit algorithm, $\sup_{\ell_1,\dots,\ell_T \in \mathcal{C}_\rho} \mathbb{E}[R_T(\ell_{1:T})] \geqslant \sqrt{T(K-1)}/504$, where the expectation is with respect to the randomness in the algorithm, and $\mathcal{C}_\rho \triangleq \left\{x \in [0,1]^K : \max_{i,j} |x_i - x_j| \leqslant \rho\right\}$.*

**Corollary 5.** *Let $K \geqslant 2$ and $T \geqslant 32(K-1)\log(14)$. Then, for any randomised bandit algorithm, there is a loss sequence $\ell_{1:T} \in [0,1]^{KT}$ such that there exists an arm $i^*$ that is optimal at every round $t$ (i.e., $\ell_{i^*,t} = \min_i \ell_{i,t}$ for all $t \geqslant 1$), but $\mathbb{E}[R_T(\ell_{1:T})] \geqslant \sqrt{T(K-1)}/504$, where the expectation is with respect to the randomness in the algorithm.*

*Proof of Corollaries 4 and 5.* Both results follow from Theorem 1 by choosing $\delta = 0.15$. Therefore there exists an $\ell_{1:T}$ such that $\mathbb{P}\{R_T(\ell_{1:T}) \geqslant \sqrt{(K-1)T\log(1/(4 \cdot 0.15)}/27\} \geqslant 0.15/2$, which implies (since $R_T(\ell_{1:T}) \geqslant 0$ here) that $\mathbb{E}[R_T(\ell_{1:T})] \geqslant \sqrt{(K-1)T}/504$. Finally note that $\ell_{1:T} \in \mathcal{C}_\rho$ since $\rho \geqslant \sqrt{(K-1)\log(1/(4\delta))/T}/(4\sqrt{\log 2})$ and there exists an $i$ such that $\ell_{i,t} \leqslant \ell_{j,t}$ for all $j$ and $t$. $\qquad \square$

## Acknowledgments

The authors would like to thank Aurélien Garivier and Émilie Kaufmann for insightful discussions. This work was partially supported by the CIMI (Centre International de Mathématiques et d'Informatique) Excellence program. The authors acknowledge the support of the French Agence Nationale de la Recherche (ANR), under grants ANR-13-BS01-0005 (project SPADRO) and ANR-13-CORD-0020 (project ALICIA).

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
