[Supplementary Material]

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

# A   Proof of Lemma 1

The proof is well known (eg., Auer et al. [2002]). We only write it for the convenience of the reader. Recall that $h_t = \big(I_s, \ell_{I_s,s}\big)_{1 \leqslant s \leqslant t-1}$. Next we write $\mathbb{Q}_j^{X|Y}$ the law of $X$ conditionally on $Y$ under $\mathbb{Q}_j$. By the chain rule for the Kullback-Leibler divergence, note that

$$\mathrm{KL}\Big(\mathbb{Q}_1^{(h_t,I_t)}, \mathbb{Q}_2^{(h_t,I_t)}\Big) = \sum_{s=1}^{t-1} \mathbb{E}_{\mathbb{Q}_1}\left[\mathrm{KL}\Big(\mathbb{Q}_1^{I_s|h_s}, \mathbb{Q}_2^{I_s|h_s}\Big) + \mathrm{KL}\Big(\mathbb{Q}_1^{\ell_{I_s,s}|(h_s,I_s)}, \mathbb{Q}_2^{\ell_{I_s,s}|(h_s,I_s)}\Big)\right]$$
$$+ \mathbb{E}_{\mathbb{Q}_1}\left[\mathrm{KL}\Big(\mathbb{Q}_1^{I_t|h_t}, \mathbb{Q}_2^{I_t|h_t}\Big)\right] . \qquad (19)$$

Note that $\mathbb{Q}_1(I_s = i|h_s) = p_{i,s} = \mathbb{Q}_2(I_s = i|h_s)$ for all $s$ (by definition of a randomised algorithm with weight vector $p_s$ at time $s$), so that the first and third KL terms equal zero. As for the second one, we can check that $\mathbb{Q}_j^{\ell_{I_s,s}|(h_s,I_s)} = Q_{j,I_s}$ (the $I_s$-th marginal of $Q_j$). Combining all these remarks with (19), we get

$$\mathrm{KL}\Big(\mathbb{Q}_1^{(h_t,I_t)}, \mathbb{Q}_2^{(h_t,I_t)}\Big) = \sum_{s=1}^{t-1} \mathbb{E}_{\mathbb{Q}_1}\big[\mathrm{KL}(Q_{1,I_s}, Q_{2,I_s})\big] = \sum_{s=1}^{t-1} \mathbb{E}_{\mathbb{Q}_1}\left[\sum_{i=1}^{K} \mathbb{1}_{\{I_s=i\}}\,\mathrm{KL}(Q_{1,i}, Q_{2,i})\right]$$
$$= \sum_{i=1}^{K} \mathbb{E}_{\mathbb{Q}_1}\big[N_i(t-1)\big]\,\mathrm{KL}\big(Q_{1,i}, Q_{2,i}\big) ,$$

which concludes the proof.

# B   Inequalities from Information Theory

We first recall below a well-known data-processing inequality that can be found, e.g., in Gray [2011, Corollary 7.2]. The main message is that transforming the data at hand can only reduce the ability to distinguish between two probability distributions.

**Lemma 5** (Contraction of entropy). *Let $\mathbb{P}$ and $\mathbb{Q}$ be two probability distributions on the same measurable space $(\Omega, \mathcal{F})$, and let $X$ be any random variable on $(\Omega, \mathcal{F})$. Denote by $\mathbb{P}^X$ and $\mathbb{Q}^X$ the laws of $X$ under $\mathbb{P}$ and $\mathbb{Q}$ respectively. Then,*

$$\mathrm{KL}\big(\mathbb{P}^X, \mathbb{Q}^X\big) \leqslant \mathrm{KL}(\mathbb{P}, \mathbb{Q}) .$$

Next we recall an inequality between the Kullback-Leibler divergence and the chi-squared divergence. In the particular case of Bernoulli distributions with parameters $p, q \in [0, 1]$, these divergences are given respectively by[1]

$$\mathrm{kl}(p, q) = p \log \frac{p}{q} + (1-p) \log \frac{1-p}{1-q} \qquad \text{and} \qquad \chi^2(p, q) = \frac{(p-q)^2}{q(1-q)} .$$

**Lemma 6** (Consequence of Lemma 2.7 in Tsybakov 2008). *Let $p, q \in [0, 1]$. Then*

$$\mathrm{kl}(p, q) \leqslant \chi^2(p, q) .$$

The final lemma is a straightforward corollary of Lemma 5 and the KL divergence between two Gaussians.

**Lemma 7.** *For $a \leqslant b$ and define $\mathrm{clip}_{[a,b]}(x) = \max\{a, \min\{x, b\}\}$. Let $Z$ be normally distributed with mean $1/2$ and variance $\sigma^2 > 0$. Define $X = \mathrm{clip}_{[0,1]}(Z)$ and $Y = \mathrm{clip}_{[0,1]}(Z - \varepsilon)$ for $\varepsilon \in \mathbb{R}$. Then $\mathrm{KL}(\mathbb{P}^X, \mathbb{P}^Y) \leqslant \varepsilon^2/(2\sigma^2)$.*

# C Proof of Theorem 4

The proof follows the same lines as that of Theorem 3. In the sequel we show that there exists a loss sequence $\ell_{1:T} \in [0,1]^{KT}$ such that $Q_T \leqslant \alpha TK$ and $\mathbb{E}[R_T(\ell_{1:T})] \geqslant (1/25)\sqrt{\alpha TK}$. As in the proof of Theorem 3, we use Lemma 4 with $\alpha/2$ and combine it with Bernstein's inequality to prove that $Q_T \leqslant \alpha TK$ with high probability.

Part 1: Applying Lemma 4 with $\alpha/2$ (note that $T \geqslant 32K \geqslant K/(4(1-\alpha/2))$ by assumption on $T$) and noting that $\max_j \mathbb{E}_{\mathbb{Q}_j}[R_T(\ell_{1:T})] \geqslant \mathbb{E}_{\bar{\mathbb{Q}}}[R_T(\ell_{1:T})]$ we get that for some $j \in \{1, \ldots, K\}$ the probability distribution $\mathbb{Q}_j$ defined with $\varepsilon = (1/2)\sqrt{(\alpha/2)(1-\alpha/2)K/T}$ satisfies

$$\mathbb{E}_{\mathbb{Q}_j}[R_T(\ell_{1:T})] \geqslant \frac{1}{8}\sqrt{\frac{\alpha}{2}\left(1-\frac{\alpha}{2}\right)TK} \geqslant \frac{1}{32}\sqrt{7\alpha TK} \tag{20}$$

since $\alpha \leqslant 1/4$ by assumption.

Part 2: Next we prove that

$$\mathbb{Q}_j(Q_T > \alpha TK) \leqslant \frac{1}{32T} \, . \tag{21}$$

To this end recall that $\mu_T = \frac{1}{T}\sum_{t=1}^{T} \ell_t$ and

$$Q_T = \sum_{t=1}^{T} \|\ell_t - \mu_T\|_2^2 = \sum_{i=1}^{K} \underbrace{\sum_{t=1}^{T}(\ell_{i,t} - \mu_{i,T})^2}_{=:v_{i,T}} \, .$$

Noting that $\ell_{i,t} \in \{0,1\}$ almost surely, we have $v_{i,T} = T\mu_{i,T}(1-\mu_{i,T}) \leqslant T\mu_{i,T} = \sum_{t=1}^{T} \ell_{i,t}$.

Recall that under $\mathbb{Q}_j$, the $\ell_{i,t}$, $t \geqslant 1$, are i.i.d. $\mathrm{Ber}(\alpha_i^j)$ where $\alpha_i^j = (\alpha/2) + \varepsilon\mathbb{1}_{\{i \neq j\}}$ (we used Lemma 4 with $\alpha/2$). We now apply Bernstein's inequality exactly as after (13): combined with a union bound, it yields that, for all $\delta \in (0,1)$, with $\mathbb{Q}_j$-probability at least $1-\delta$, for all $i \in \{1, \ldots, K\}$,

$$\sum_{t=1}^{T} \ell_{i,t} \leqslant T\alpha_i^j + \sqrt{2T\alpha_i^j\left(1-\alpha_i^j\right)\log\frac{K}{\delta}} + \frac{1}{3}\log\frac{K}{\delta}$$

$$\leqslant T\left(\frac{\alpha}{2}+\varepsilon\right) + \sqrt{2T\left(\frac{\alpha}{2}+\varepsilon\right)\log\frac{K}{\delta}} + \frac{1}{3}\log\frac{K}{\delta} \, . \tag{22}$$

Now note that, by definition of $\varepsilon = (1/2)\sqrt{(\alpha/2)(1-\alpha/2)K/T}$ and by the assumption $T \geqslant 8K/\alpha$,

$$\frac{\alpha}{2}+\varepsilon \leqslant \frac{\alpha}{2}+\frac{1}{2}\sqrt{\frac{\alpha K}{2T}} \leqslant \frac{5\alpha}{8} \, .$$

Substituting the last upper bound in (22) and using the assumption $T\alpha \geqslant 4\log(K/\delta)$ (that we check later) to obtain $\log(K/\delta) \leqslant (1/2)\sqrt{T\alpha\log(K/\delta)}$ we get

$$\sum_{t=1}^{T} \ell_{i,t} \leqslant \frac{5T\alpha}{8} + \left(\frac{\sqrt{5}}{2}+\frac{1}{6}\right)\sqrt{T\alpha\log\frac{K}{\delta}} \leqslant \frac{5T\alpha}{8} + \frac{3T\alpha}{8} = T\alpha \, , \tag{23}$$

where the last inequality is true whenever $T\alpha \geqslant c_1\log(K/\delta)$ with $c_1 = (4/9)^2(3\sqrt{5}+1)^2$. By the assumption $\alpha \geqslant 2c_1\log(T)/T \geqslant c_1\log(32TK)/T$ (since $T \geqslant 32K$), the condition $T\alpha \geqslant c_1\log(K/\delta)$ is satisfied for $\delta = 1/(32T)$ (as well as the weaker condition $T\alpha \geqslant 4\log(K/\delta)$ mentioned above). We conclude the proof of (21) via $Q_T = \sum_{i=1}^{K} v_{i,T} \leqslant \sum_{i=1}^{K}\sum_{t=1}^{T} \ell_{i,t} \leqslant \alpha TK$ by (23).

Conclusion: We show by contradiction that there exists a loss sequence $\ell_{1:T} \in [0,1]^{KT}$ such that $Q_T \leqslant \alpha TK$ and

$$\mathbb{E}[R_T(\ell_{1:T})] \geqslant \frac{1}{64}\sqrt{7\alpha TK} \, , \tag{24}$$

where the expectation is with respect to the internal randomisation of the algorithm. Imagine for a second that (24) were false for every loss sequence $\ell_{1:T} \in [0,1]^{KT}$ satisfying $Q_T \leqslant \alpha TK$. Then we would have $\mathbb{1}_{\{Q_T \leqslant \alpha TK\}} \mathbb{E}_{\mathbb{Q}_j}[R_T(\ell_{1:T})|\ell_{1:T}] \leqslant (1/64)\sqrt{7\alpha TK}$ almost surely (since the internal source of randomness of the bandit algorithm is independent of $\ell_{1:T}$). Therefore, using the tower rule for the first expectation on the r.h.s. below, we would get

$$
\begin{aligned}
\mathbb{E}_{\mathbb{Q}_j}[R_T(\ell_{1:T})] &= \mathbb{E}_{\mathbb{Q}_j}\big[R_T(\ell_{1:T})\mathbb{1}_{\{Q_T \leqslant \alpha TK\}}\big] + \mathbb{E}_{\mathbb{Q}_j}\big[R_T(\ell_{1:T})\mathbb{1}_{\{Q_T > \alpha TK\}}\big] \\
&\leqslant \frac{1}{64}\sqrt{7\alpha TK} + T \cdot \mathbb{Q}_j(Q_T > \alpha TK) \\
&\leqslant \frac{1}{64}\sqrt{7\alpha TK} + \frac{1}{32} < \frac{1}{32}\sqrt{7\alpha TK}
\end{aligned}
\tag{25}
$$

where (25) follows from (21) and by noting that $1/32 < (1/64)\sqrt{7\alpha TK}$ since $\alpha \geqslant 8K/T > 4/(7TK)$. Comparing (25) and (20) we get a contradiction, which proves that there exists a loss sequence $\ell_{1:T} \in [0,1]^{KT}$ satisfying both $Q_T \leqslant \alpha TK$ and (24). We conclude the proof by noting that $\sqrt{7}/64 \geqslant 1/25$.

Nota: the assumption $T \geqslant (32K) \vee 601$ is sufficient to make the interval $\left[\frac{2c_1 \log(T) \vee (8K)}{T}, \frac{1}{4}\right]$ non empty.

## D  Proof of Lemma 3

First we use the definition of the losses to bound

$$
R_T(\ell_{1:T}^i) = \sum_{t=1}^{T}(\ell_{I_t,t} - \ell_{i,t}) \geqslant \Delta \sum_{t=1}^{T} \mathbb{1}_{\{Z_t \in [2\Delta, 1-2\Delta] \text{ and } I_t \neq i\}} .
$$

Let $W_t = \mathbb{1}_{\{Z_t \in [2\Delta, 1-2\Delta]\}}$, which forms an i.i.d. Bernoulli sequence with

$$
\mathbb{Q}_i(W_t = 0) \leqslant \exp\left(-\frac{(1/2 - 2\Delta)^2}{2\sigma^2}\right) \triangleq p \leqslant 1/8 ,
$$

where the inequality follows by standard tail bounds on the Gaussian integral [Boucheron et al., 2013, Exercise 2.7]. Therefore by Hoeffding's bound

$$
\mathbb{Q}_i\left(\sum_{t=1}^{T} W_t \leqslant \frac{3T}{4}\right) = \mathbb{Q}_i\left(\sum_{t=1}^{T} W_t - T\mathbb{E}_{\mathbb{Q}_i}[W_1] \leqslant \frac{3T}{4} - (1-p)T\right) \leqslant \exp(-T/32) \leqslant \delta/2 .
$$

The first part of the statement follows from the union bound and because if $N_i(T) \leqslant T/2$ and $\sum_{t=1}^{T} W_t \geqslant 3T/4$, then

$$
\Delta \sum_{t=1}^{T} \mathbb{1}_{\{Z_t \in [2\Delta, 1-2\Delta] \text{ and } I_t \neq i\}} \geqslant \Delta T/4 .
$$

For the second part we use below the tower rule (conditioning on $I_t$ and the history $h_t$ up to $t-1$) and the fact that $W_t$ is independent of $I_t$ given $h_t$ but also independent of $h_t$ to get that

$$
\begin{aligned}
\mathbb{E}_{\mathbb{Q}_i}[R_T(\ell_{1:T}^i)] &\geqslant \Delta \sum_{t=1}^{T} \mathbb{Q}_i(W_t = 1 \text{ and } I_t \neq i) \\
&\geqslant \Delta \sum_{t=1}^{T} \mathbb{Q}_i(W_t = 1)\, \mathbb{Q}_i(I_t \neq i) \geqslant \frac{7\Delta}{8}\mathbb{E}_{\mathbb{Q}_i}[T - N_i(T)] ,
\end{aligned}
$$

which completes the proof.

# E Proof of Corollary 3

Suppose on the contrary that such a strategy exists. Then

$$\mathbb{E}[R_T(\ell_{1:T})] \leqslant \int_0^\infty \mathbb{P}\left(R_T(\ell_{1:T}) \geqslant x\right) dx$$

$$\leqslant \int_0^\infty \exp\left(-\left(\frac{x}{C\sqrt{(K-1)T}}\right)^{\frac{1}{p}}\right) dx \leqslant C\sqrt{(K-1)T}.$$

By the assumption in the corollary we have

$$\delta \geqslant \mathbb{P}\left(R_T(\ell_{1:T}) \geqslant C\sqrt{(K-1)T}\log^p(1/\delta)\right)$$

$$= \mathbb{P}\left(R_T(\ell_{1:T}) \geqslant \frac{\sqrt{(K-1)T}\log(1/(16\delta))}{203C} \cdot \frac{203C^2\log^p(1/\delta)}{\log(1/(16\delta))}\right),$$

which leads to a contradiction by choosing $\delta$ sufficiently small and $T$ sufficiently large and applying Theorem 2 to show that there exists an $\ell_{1:T} \in [0,1]^{KT}$ for which

$$\mathbb{P}\left(R_T(\ell_{1:T}) \geqslant \frac{\sqrt{(K-1)T}\log(1/(16\delta))}{203C}\right) \geqslant 2\delta.$$