[Reviews · NeurIPS 2016]

Reviewer 1

Summary

This paper considers a setting of adversarial multi-armed bandits, when the learner must compete against the best constant pulling strategy in hindsight. The paper studies achievable lower bounds in this setting. Three types of lower bounds are considered. "Zeroth order" lower bounds in high probability, that depend on the time horizon T: Theorem 1, 2, Corollary 3. See also Corollary 4, 5. "First order" lower bounds, where the time horizon is replaced with the cumulative loss of the optimal action L*_T: Theorem 3. "Second order" lower bounds, where the time horizon is replaced with the quadratic variation of the losses: Theorem 4. The paper discusses consequences of these lower bounds. Regarding Zeroth order bounds, they imply that existing algorithm are close to optimal up to a sqrt(log(K)) factor, they also imply that no algorithm can exploit information such as the range of the losses (is sufficiently high), or that one action is optimal at all steps.

Qualitative Assessment

I have read the feedback from the authors. I think this is an interesting contribution, and I vote for acceptance, but you should highlight the novelty much more. My assessment regarding novelty was to make you aware that it should be highlighted. ** Not many papers focus on lower bounds. This contribution is remarkable in this sense. As mentioned, the results are not very surprising, and many algorithms already achieve near-optimal results. Also, the novel results do not seem to give much more insights about the design of algorithms. *** Technical Quality: Good. Results look sound. Novelty/Originality: None. Potential impact or usefulness: Low. Clarity and presentation: Very good.

Confidence in this Review

3-Expert (read the paper in detail, know the area, quite certain of my opinion)


Reviewer 2

Summary

The authors consider the K-armed bandit in the adversarial setting, where the loss of each arm at each time is in [0,1] and chosen by an adversary. The authors consider several refinements of the classical setting (e.g. that of Auer 2002) as follows: - when the loss of the best action is small with respect to T - when the quadratic variation of the losses - when the regret must be bounded in high probability (rather than in expectation). Probability is with respect to the algorithm's randomization. The authors provide regret lower bounds that apply to any algorithm, and compare them with available upper bounds.

Qualitative Assessment

The results look good overall. The lower bound provided here make a nice contribution to the large body of litterature on adversarial bandits. In particular one of the proofs involves selecting loss sequences such that the optimal arm is the same across all rounds, which is surprising (showing the gap between full-information and bandit). The authors further open an interesting problem which is how the gap between the lower and upper bound (Hazan and Kale 2011) may be closed in the "small variance" setting. While the general proof techniques are standard (using inequalities between the Kullback divergence and other statistical distances (chi-square, total variation etc.), the proofs are not trivial and involve carefully chosen distributions on the sequence of losses.

Confidence in this Review

2-Confident (read it all; understood it all reasonably well)


Reviewer 3

Summary

This paper presents several lower bounds for adversarial bandit setting. In most cases, upper bounds matching these lower bounds (within logarithmic factors) are known, but it was not known earlier if those upper bounds bounds were tight. In one case (second order bounds in terms quadratic variation) there is a gap between the best known upper bound and the lower bound shown by the authors.

Qualitative Assessment

Merits: It is a good paper to read, well-written and with very few typos. While most results provided in this paper are as one would expect in these problems (the authors also note this), I think it does the necessary job of filling the gaps in this literature and prove tightness of many existing algorithms. From technical point of view, the construction of stochastic correlated losses used for Theorem 1 and 2 (zero order bounds) and its proof seems particularly interesting, and different from the common constructions for lower bounds in this literature (like the one used in Section 3 for first order bounds). It will be useful to provide some further intuition on why constructions like those used first order bounds don't work for zero order bounds. I can see that Section 3 techniques are good for proving expected bound, while for zero order bounds authors need a construction where they can employ Lemma 2. But, the authors may be able to provide a better high level intuition. Critique: As mentioned earlier, the results in these papers are not very surprising. Most of the techniques, barring the construction Section 2, are standard in the literature. Summary of review: Overall, I think it is a good technical paper, which will be of interest and useful to researchers working in this area.

Confidence in this Review

3-Expert (read the paper in detail, know the area, quite certain of my opinion)


Reviewer 4

Summary

The paper provides new lower bounds on the regret that must associated with adversarial bandit algorithms for three cases ("classes"): 1) It is shown that here is little room for improvement over existing strategies, for the regret achievable with high probability. 2) Given some additional regularity conditions (“nice” data) the paper establishes lower bounds that replace the horizon T in the regret bound with the loss of the best action ("first-order bounds"), and 3) also with the quadratic variation of the losses ("second-order bounds").

Qualitative Assessment

The results are interesting. The paper however appears to have been written in a harry and clarity and rigor suffer, e.g., I could not find a discussion on the impact of the assumption on "the existence of a single arm that is optimal in every round" in any place other than in the abstract...

Confidence in this Review

3-Expert (read the paper in detail, know the area, quite certain of my opinion)


Reviewer 5

Summary

Paper proves a variety of fine-grained lower bounds for adversarial bandits. Aside from the second order bounds, the lower bounds match known upper bounds up to sqrt(log(K)) factors. In addition they make connections to impossibility results for bandits motivated by performance guarantees in the full information case.

Qualitative Assessment

Aside from the so-called "fatal flaw" commented on above (I write this under the assumption that the paper can be "revived" in the rebuttal) this paper clearly described the problems they were after, their contributions, and the machinery from past art used to prove the results along with their novel constructions. The proofs and the proof sketches were very clear -------- After Rebuttal period ---------- I agree that the bug is fixable.

Confidence in this Review

3-Expert (read the paper in detail, know the area, quite certain of my opinion)


Reviewer 6

Summary

This paper focuses on three classes of lower bound for adversarial bandits-- zero order, first order and second order lower bound. First of all, the paper proves two new high-probability zero order lower bounds. The first shows that no strategy can enjoy smaller regret than $\Omega(\sqrt{KT log(1/\delta)})$ w.p. at least $1-\delta$. The second shows that any algorithm whose regret is $O(\sqrt{KT})$ must necessarily suffer a high probability regret of at least $\Omega(\sqrt{KT} log(1/\delta))$. In the section of first order lower bound, the paper gives a new lower bound of $\sqrt{L_T^*K}$. Similarly, in the section of second order lower bound, the paper gives a lower bound of $\sqrt{Q_T K}$, where $Q_T$ is the quadratic variation of the loss sequence $l_t$.

Qualitative Assessment

The paper mainly addresses the high-probability zero order bound, and first order, second order lower bound for the adversarial bandits, as well as two impossibility result in bandit setting, which is possible both in full-information setting. The zero order high-probability bound shows that there is little room(\sqrt{log K}) for the known best bandit algorithm to improve. The construction in the proof has the nice property that optimal arm is the same in every round and that the range of losses in each round is $O(\sqrt{K\log(1/\delta)/T})$, which is useful in the impossibility results and give people insight in future research. The proofs of first-order and second-order lower bound are basically based on the same construction, inspired by Auer et. al 2002. The paper is organized well and written in high quality with no typo found, which makes it a pleasure to read. The proofs are sound and easy to understand.

Confidence in this Review

2-Confident (read it all; understood it all reasonably well)